

# Morphological grounds for the obligate aerial respiration of an aquatic snail: functional and evolutionary perspectives

Cristian Rodriguez[1,2,3,*], Guido I. Prieto[3,4,*], Israel A. Vega[1,2,3] and Alfredo Castro-Vazquez[1,2,3]

[1] IHEM, CONICET, Universidad Nacional de Cuyo, Mendoza, Argentina
[2] Instituto de Fisiología, Facultad de Ciencias Médicas, Universidad Nacional de Cuyo, Mendoza, Argentina
[3] Departamento de Biología, Facultad de Ciencias Exactas y Naturales, Universidad Nacional de Cuyo, Mendoza, Argentina
[4] Department of Philosophy I, Ruhr University Bochum, Bochum, Germany
* These authors contributed equally to this work.

Corresponding authors
Israel A. Vega,
israel.vega7@gmail.com
Alfredo Castro-Vazquez,
a.castrovazquez@gmail.com

## ABSTRACT

The freshwater caenogastropod family Ampullariidae is emerging as a model for a variety of studies, among them, the evolution of terrestriality. A common character of the family is that all its members bear a lung while retaining the ancestral gill. This ensures that many ampullariids are able to inhabit poorly oxygenated waters, to bury in the mud during estivation, and to temporarily leave the water, in some species for oviposition. To these characters *Pomacea canaliculata* (Caenogastropoda, Ampullariidae) adds that is an obligate air-breather. In a recent paper, we showed the gill epithelium of *P. canaliculata* has a set of characteristics that suggest its role for oxygen uptake may be less significant than its role in ionic/osmotic regulation and immunity. We complement here our morphological investigation on the respiratory organs of *P. canaliculata* by studying the lung of this species at the anatomical (3D reconstructions of the blood system and nerve supply), histological and ultrastructural levels. The circulation of the gill and the lung are interconnected so that the effluence of blood from the gill goes to the lung where it completes oxygenation. Besides that, we found the lung cavity is lined by a pavement epithelium that encloses an anastomosing network of small blood spaces resting over a fibromuscular layer, which altogether form the respiratory lamina. The pavement cells form a blood-gas barrier that is 80–150 nm thick and thus fulfils the requirements for an efficient gas exchanger. Tufts of ciliary cells, together with some microvillar and secretory cells, are interspersed in the respiratory lamina. Rhogocytes, which have been proposed to partake in metal depuration and in the synthesis of hemocyanin in other gastropods, were found below the respiratory lamina, in close association with the storage cell tissue. In light of these findings, we discuss the functional role of the lung in *P. canaliculata* and compare it with that of other gastropods. Finally, we point to some similarities in the pattern of the evolution of air dependence in this family.

## INTRODUCTION

Oxygen uptake by an animal is easier in air than in water, because of the lower viscosity and the higher concentration and diffusion rate of oxygen in the air. Thus, it is not surprising that air respiration has evolved many times among the Metazoa (*Maina, 2014*; *Schmidt-Nielsen, 1997*).

Among the Mollusca, the primitive respiratory organ may have been the mantle cavity (*Haszprunar, 1992*), which would have provided enough surface for gas exchange for the small-sized primitive molluscs (*Runnegar & Pojeta, 1985*). Later, however, some gastropods have adapted to air-breathing by modifying the mantle cavity (or a part of it) as a lung. In most of these cases, the modifications consisted of the vascularization of part of the mantle cavity roof and specializations of the inner mantle epithelium for gas exchange (*Hyman, 1967*), and these modifications for air-breathing were often accompanied by the reduction or loss of gills (*Ponder, Lindberg & Ponder, 2019*).

However, the caenogastropod family Ampullariidae is peculiar in that it comprises species that have retained the ancestral gill together with the evolutionary acquisition of a lung that results from a secondary invagination of the roof of the mantle cavity (*Brooks & McGlone, 1908*). The primary one is the one that results in the mantle cavity itself, which remains connected with the adult lung through a contractile pneumostome. The family includes fully-aquatic species that lay gelatinous egg masses underwater, as well as others that depend on aerial respiration to lay calcareous eggs above water level (*Hayes et al., 2009*) or to remain buried in the mud during long estivation periods (*Giraud-Billoud et al., 2011*). Among the latter, *Pomacea canaliculata* (Lamarck, 1822) adds that is an obligate air-breather (*Seuffert & Martín, 2009*; *Seuffert & Martín, 2010*).

A recent comparative report of the genomes of four ampullariid species (*P. canaliculata*, *Pomacea maculata*, *Marisa cornuarietis* and *Lanistes nyassanus*) has disclosed signatures of terrestrialization among them (*Sun et al., 2019*). In both *Pomacea* species, two gene families appear expanded and are related to the aerial oviposition that is typical of the genus. One is related to the synthesis of a calcium-binding protein involved in egg-capsule formation, and the other to the codification of a complex neurotoxin present in eggs, which is a deterrent for egg predation (*Dreon et al., 2013*; *Hayes et al., 2015*).

As mentioned above, the lung is also a hallmark of the family. Its gross anatomy has attracted the interest of researchers since the 19th century (*Andrews, 1965*; *Bavay, 1873*; *Berthold, 1991*; *Bouvier, 1891*; *Jourdain, 1879*; *Prashad, 1925*; *Sabatier, 1879*; *Starmühlner, 1964*), but this organ was initially interpreted as a gas bladder for controlling the animal's buoyancy and its role as a respiratory organ was recognized only later (*Quoy & Gaimard, 1833*). Since these studies were based on dissections and observations of the living animal, it is not surprising that just two studies were successful in identifying

the lung microstructures specifically involved in gas exchange, namely those of *Lutfy & Demian (1965)* in *Marisa cornuarietis* and of *Rodriguez et al. (2018)* in *P. canaliculata*. In the latter study, we proposed the occurrence of a "respiratory lamina", composed of adjacent and interconnected blood spaces (*Rodriguez et al., 2018*), which would allow blood to be in close proximity with air, thus enabling an efficient gas exchange. Besides those studies, *Mueck, Deaton & Lee (2020b)* have recently studied similar structures in *Pomacea maculata* under the electron microscope, but they have erroneously equated the respiratory epithelium with the tall, microvillar and pigmented outer mantle epithelium, which is the external cover of the lung roof.

Therefore, an important focus of the current work has been the fine structure of the respiratory lamina, the proposed gas-exchange barrier (*Rodriguez et al., 2018*), but we also studied the lung blood and nerve supply. Particularly, we studied the circulatory connections that would explain why the oxygenation of the blood is completed in the lung, even when the gill may not be functioning. We provide a thorough description of the lung of *P. canaliculata* based on light and electron microscopy and computerized 3D reconstructions. We report the unexpected finding of rhogocytes, for the first time in an ampullariid. Finally, we discuss our results from a functional and evolutionary perspective.

## MATERIALS AND METHODS

### Animals and dissections

Animals (young-adult males, 20 mm in shell length) from our cultured Rosedal strain (see *Rodriguez et al., 2019* for details) were kept in a water bath at 4°C for 20–30 min to provoke relaxation and minimize nociception. After carefully cracking the shell, we dissected out the lung and surrounding tissues both for histological examinations and 3D reconstruction of the blood and nerve systems of the lung.

We dissected out lung sacs for routine light microscopy (from $N = 6$ animals); samples that contained most of the mantle edge and the copulatory apparatus, the lung, as well as parts of the gill, ureter, and pericardium, for 3D reconstructions (from $N = 2$ animals); and pieces of the lung floor and roof for scanning and transmission electron microscopy (SEM and TEM, from $N = 4$ and $N = 10$ animals, respectively).

The Institutional Committee for the Care and Use of Laboratory Animals (Comité Institucional para el Cuidado y Uso de Animales de Laboratorio, Facultad de Ciencias Médicas, Universidad Nacional de Cuyo) approved the procedures for animal culture, sacrifice and tissue sampling (approval protocol N° 55/2015).

### Light microscopy and 3D reconstruction of the lung's blood system and innervation

The histological processing involved sample fixation in dilute (1:2) Bouin's fluid, dehydration in a graded ethanol series, clearing in xylene, paraffin-embedding (Histoplast, Argentina), and sectioning (5–10 μm-thick) on a rotary microtome (Microm). After staining with Gill's hematoxylin-eosin, we examined and photographed the samples under a Nikon Eclipse 80i microscope using Nikon DS-Fi1-U3 camera and Nikon NIS-ELEMENT Image Software.

Also, we examined serially sectioned (10 μm-thick) Harris hematoxylin-eosin-stained lung samples under a Nikon Alphaphot-2 YS2 microscope equipped with a Nikon Digital Sight DS-5M camera. We selected the images of every fifth section for the 3D-reconstruction of the blood system and innervation of the lung. For this purpose, we used the software Reconstruct 1.1.0.0 (*Fiala, 2005*) and followed the procedure described by *Ruthensteiner & Heß (2008)* to assemble the final PDF-model, as described in *Rodriguez et al. (2019)*.

### Scanning and transmission electron microscopy

For SEM examination, we cut-opened the lung sacs to wash the lung cavity out of mucus, and then we fixed them with dilute Bouin's fluid. After that, sample processing involved dehydration in a graded acetone series, critical point-drying, mounting on aluminum stubs, coating with gold, and examination with a Jeol/EO JSM-6490LV electron microscope.

The preparation of both lung roof and floor samples for TEM involved fixation in Karnovsky's fluid, post-fixation in 1% osmium tetroxide overnight, dehydration in a graded acetone series, and embedding in Spurr's resin. We obtained silver-grey, ultrathin sections, which we mounted on copper grids, stained them with uranyl acetate and lead citrate, and examined them with a Zeiss EM 900 microscope. Besides, we used ~200 nm-thick ultramicrotome sections stained with toluidine blue and mounted with DPX medium (Cat. #44581; Sigma–Aldrich, Sigma–Aldrich, St. Louis, MO, USA) for topographical orientation and examination under light microscopy.

## RESULTS

### General organization of the lung

The lung of *P. canaliculata* (Fig. 1A) is a single sac that occupies most of the roof of the mantle cavity. It abuts the gill on its posterior and right margins. An opening in the lung floor, namely the pneumostome, lies anteriorly, near the osphradium and the left mantle edge, and communicates the lung cavity with the mantle cavity. The pneumostome shows two lips (anterior and posterior) and can close by their apposition. Prominent blood sinuses traverse both the lung roof and floor. The muscular layer of the lung floor is formed of muscle strands arranged in two directions, thus is thicker than that of the roof, particularly in the region rostral to the pneumostome.

From dorsal to ventral, the lung roof (Fig. 1B) is composed of (1) the outer mantle epithelium, formed by columnar pigment and mucous cells; (2) a single muscle layer; (3) a vascular layer containing the main blood sinuses supplying the lung, which are surrounded by the perivascular storage tissue (*Giraud-Billoud et al., 2011*); and (4) the respiratory lamina. Almost mirroring that arrangement, the lung floor (Fig. 1C) is composed, from dorsal to ventral, of (1) the respiratory lamina; (2) the vascular and storage tissue layer, which is thicker than that of the roof; (3) a double muscle layer with fibers arranged in perpendicular directions; and (4) the inner mantle epithelium that lines the mantle cavity, and which is formed by columnar ciliated and mucous cells.

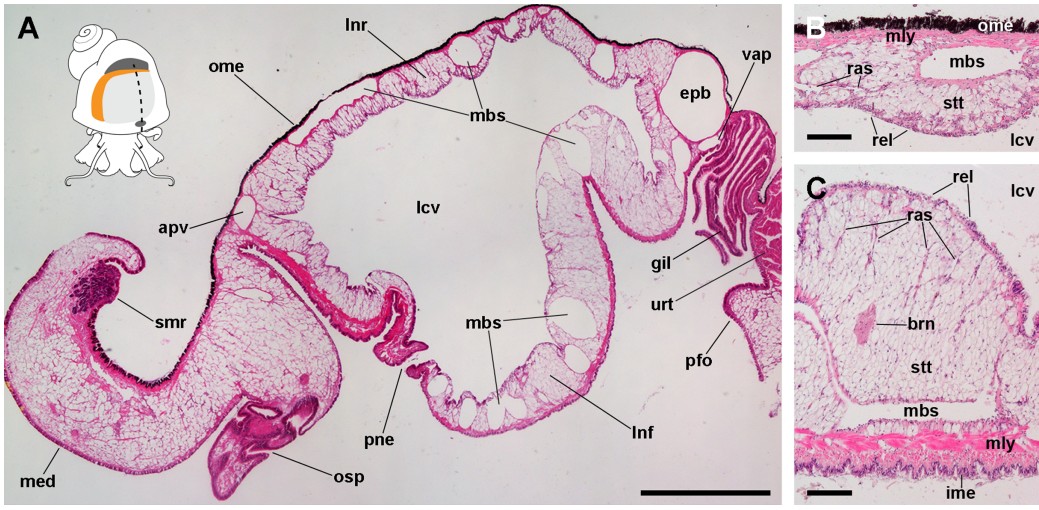

**Figure 1 The lung and the left pallial organs.** (A) The lung and related pallial structures (parasagittal section, as indicated in the thumbnail sketch). (B) The roof of the lung. (C) The floor of the lung. Abbreviations: apv, afferent pulmonary vein; brn, branchial nerve; epb, efferent pulmobranchial vein; gil, gill; ime, inner mantle epithelium; lcv, lung cavity; lnf, lung floor; lnr, lung roof; mbs, main blood sinuses; med, mantle edge; mly, muscle layer; ome, outer mantle epithelium; osp, osphradium; pfo, pallial fold; pne, pneumostome; ras, radial sinuses; rel, respiratory lamina; smr, supramarginal ridge; stt, storage tissue; urt, ureter; vap, ventral afferent pulmonary vein. Hematoxylin and eosin. Scale bars represent: (A) 1 cm; (B) 50 μm; (C) 100 μm.

## Blood supply and innervation of the lung

The 3D reconstruction of the lung and related pallial organs showed the blood supply to the roof and the floor comes from different origins (Fig. 2; see the interactive 3D model in Fig. S1). The *afferent pulmonary vein*, which is a prolongation of the *afferent branchial vein*, runs along the anterior and left margins of the lung, branching repeatedly and supplying the lung roof all along its course. Upon irrigating the lung roof, the blood reaches the *efferent pulmobranchial vein*, which conveys blood also from the gill leaflets and carries it to the heart auricle (Figs. 2B and 2E).

 In turn, the lung floor receives blood from several origins and, for clarity, will be treated separately as the right (Fig. 2C) and the left afferent systems (Fig. 2D). Connection of the gill and lung circulation occurs through the *right afferent sinuses*. The *ventral afferent pulmonary vein* is the main right afferent and runs parallel to the already mentioned *efferent pulmobranchial vein*. It receives blood from the gill leaflets and divides within the right half of the lung floor into numerous *right afferent sinuses* (Figs. 2C and 2F). All the latter will end at the *ventral efferent pulmonary vein*, which follows a diagonal path within the lung floor. This large vein arises as a branch of the *efferent pulmobranchial vein*, and finally joins it again close to its entrance into the auricle. Also, a branch arising from the *afferent pulmonary vein*, near the osphradium, turns to the right of the snail and ramifies into the anterior part of the lung floor. The latter branches will join the *efferent pulmobranchial vein* (Figs. 2C and 2F).

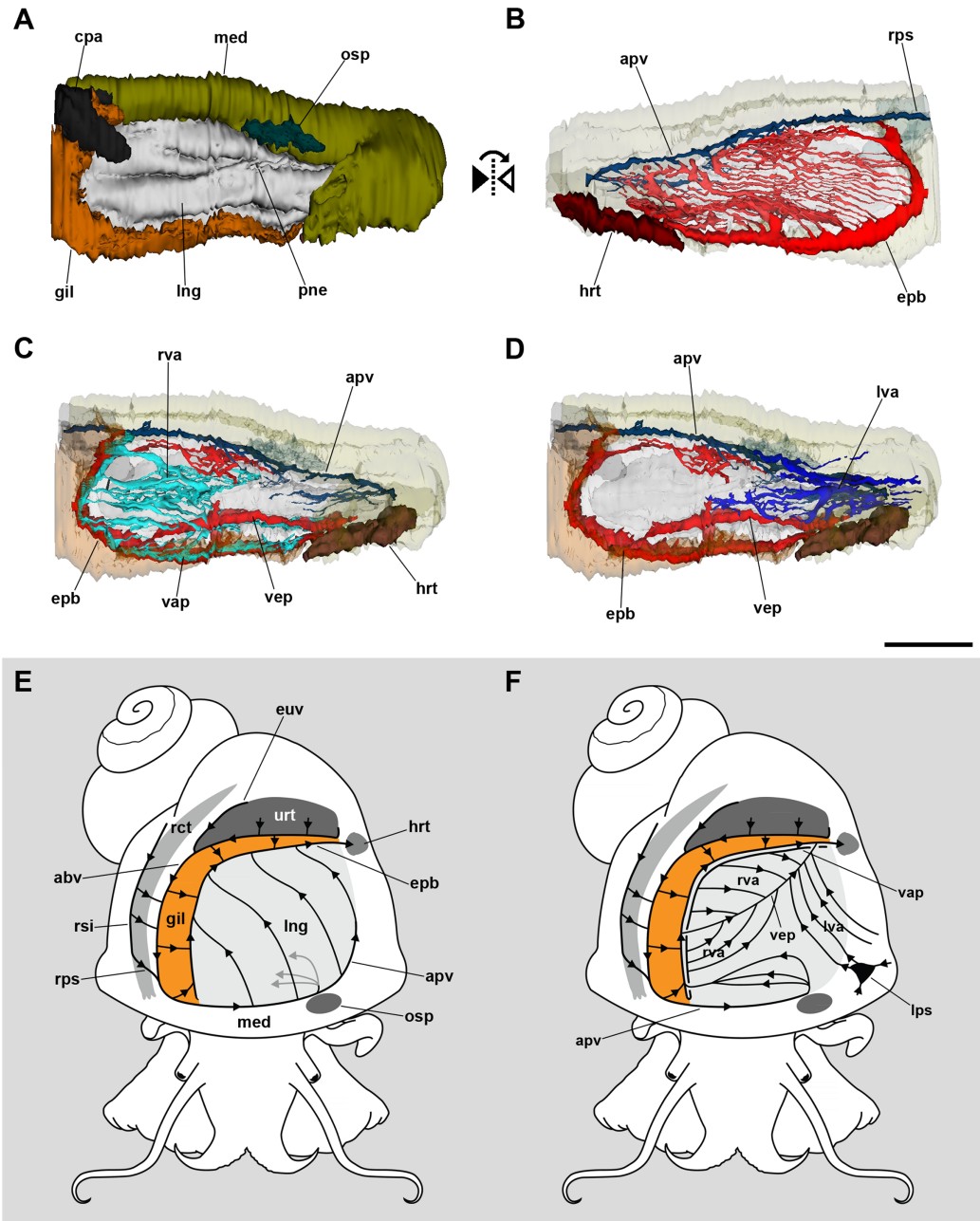

**Figure 2 Computerized 3D reconstruction of the lung vasculature.** (A) Ventral view of the mantle cavity showing the lung abutting the ctenidium and the mantle edge. (B) Dorsal view of the rather parallel thin sinuses that irrigate the lung roof. (C) Ventral view of the right vascular system in the floor of the lung. (D) Ventral view of the left vascular system in the floor of the lung. (E) Diagram of the blood flow in sinuses of the lung roof. (F) Diagram of the blood flow in sinuses of the lung floor. All blood from the lung's roof and floor converges into the efferent pulmobranchial vein, which conveys it to the heart auricle. Abbreviations: abv, afferent branchial vein; apv, afferent pulmonary vein; cpa, copulatory apparatus; epb, efferent pulmobranchial vein; euv, efferent ureteral vein; gil, gill; hrt, heart; lng, lung; lps, left pallial sinus; lva, left ventral afferent sinuses; med, mantle edge; osp, osphradium; pne, pneumostome; rct, rectum; rps, right pallial sinus; rsi, rectal sinus; rva, right ventral afferent sinuses; urt, ureter; vap, ventral afferent pulmonary vein; vep, ventral efferent pulmonary vein. Scale bar represents 1 mm.

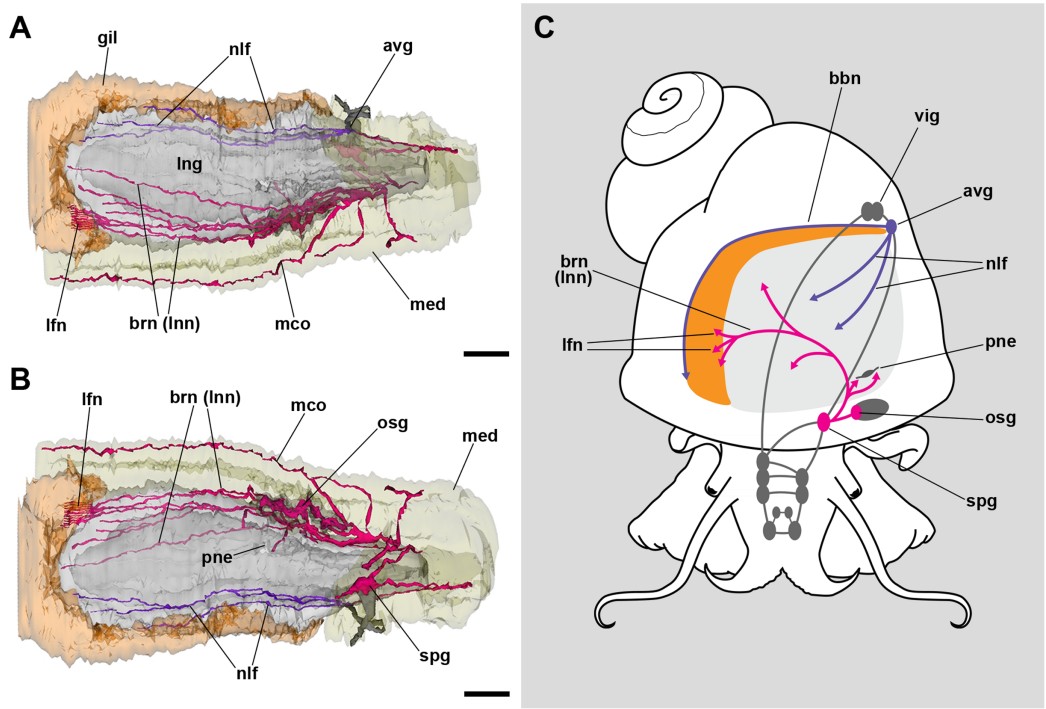

**Figure 3 Computerized 3D reconstruction of the lung innervation.** (A) Dorsal view of the innervation in the lung roof. (B) Ventral view of the innervation in the lung floor. (C) Diagram showing the major ganglia (grey), the nerves, and the accessory ganglia providing the lung (pale grey) and gill (orange) innervation. The lung is innervated from the supraesophageal ganglion by branches of the branchial nerve (pink), which also innervate the pneumostome. Innervation in the floor of the lung originates from an accessory visceral ganglion located at the pericardial wall (violet). Abbreviations: avg, accessory visceral ganglion; bbn, branchial base nerve; brn (=lnn), branchial and lung nerves; gil, gill; lfn, branchial leaflet nerve; lng, lung; mco, mantle commisure; med, mantle edge; nlf, nerves of the lung floor; osg, osphradial ganglion; pne, pneumostome; spg, supraesophageal ganglion; vig, visceral ganglion.

The left part of the lung floor is irrigated by numerous *left afferent sinuses* that arise from left pallial sinuses (Figs. 2D and 2F). All these *left afferent sinuses* end in the *ventral efferent pulmonary vein*. Finally, the two largest efferent veins (the *ventral efferent pulmonary vein* and the *efferent pulmobranchial vein*) merge upon entering the auricle (Figs. 2D and 2F).

The lung roof is innervated by branches arising from the *supraesophageal ganglion* while the floor is innervated from the *accessory visceral ganglion* (Fig. 3; Fig. S1). In addition, the *supraesophageal ganglion* sends two branches to the pneumostome. The smaller nerve branches could not be followed in hematoxylin–eosin preparations, but neurites were found spread in TEM preparations of the respiratory lamina, which will be described below.

## The inner surface of the lung and the pneumostome

Under SEM, the lung cavity appears covered by pavement cells, with irregularly interspersed tufts of ciliated columnar cells (Fig. 4A). Numerous microvilli occur in the

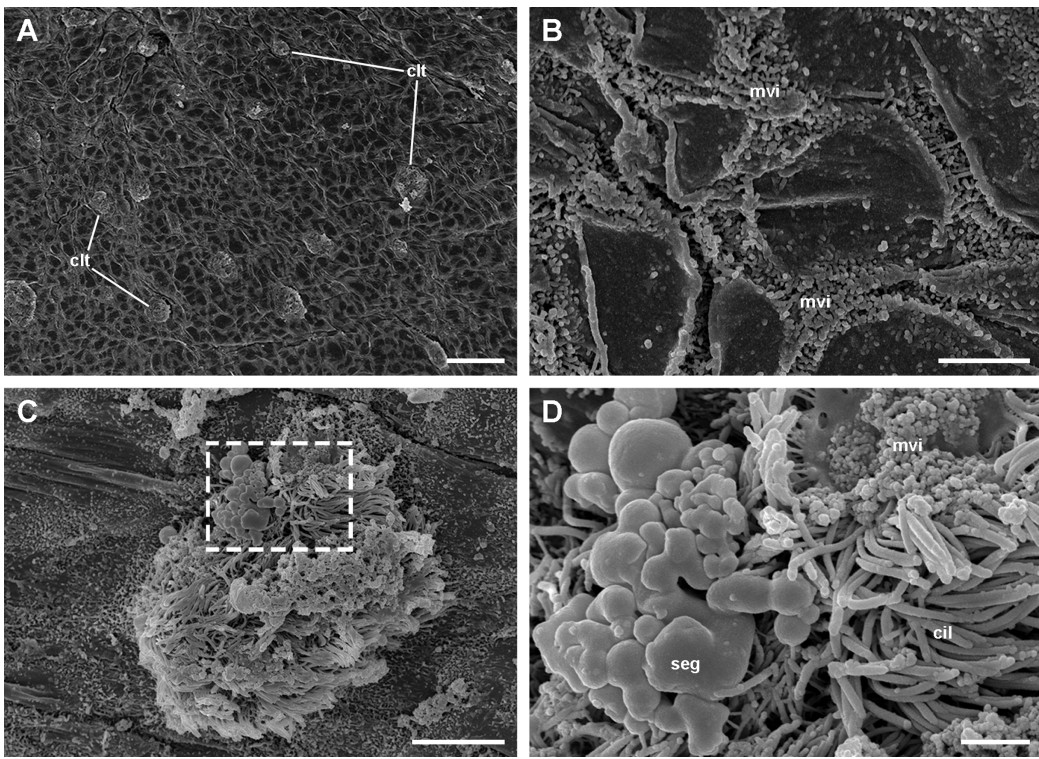

**Figure 4 The lung cavity.** (A) The surface of the lung cavity shows ciliary tufts irregularly interspersed amongst the pavement cells covering the blood sinus of the respiratory lamina. (B) The grooves between the pavement cells show short and numerous microvilli. (C and D) Ciliary tufts. (C) Different types of apical specialisations (dashed box; magnified in D): merging secretory globules, long and numerous cilia, and short microvilli. Abbreviations: cil, cilia; clt, ciliary tufts; mvi, microvilli; seg, secretory globules. Scanning electron microscopy. Scale bars represent: (A) 50 µm; (B) 5 µm; (C) 10 µm; (D) 2 µm.

shallow grooves between pavement cells and some of them show enlarged tips (Fig. 4B); occasionally, pavement cells show microvilli on its central surface too, but these are shorter than those found in the grooves. The ciliary tufts (Fig. 4C) show cells with different apical specializations besides their long cilia, such as short microvilli and partially fused secretory globules (Fig. 4D). These apical specializations are further described in the corresponding section, as seen under TEM.

The pneumostome communicates the lung and the mantle cavities and an epithelium of ciliated and mucous cells lines it, which is continuous with the epithelium lining the rest of the mantle cavity (see Fig. 1C). In contrast, the epithelium covering the lung cavity side of the pneumostome lips (Fig. 5) shows ciliated cells arranged in single or double parallel rows, which gives the lips a striated appearance (Figs. 5A and 5B). Around this region of parallel rows, there are fields of densely packed ciliated cells (Fig. 5C) which are in turn surrounded by pavement cells. These cells are not separated by grooves (as in the respiratory lamina), but by ciliary rows (Fig. 5D). This zone marks the transition to the typical pavement of the respiratory lamina (Figs. 4A and 4B).
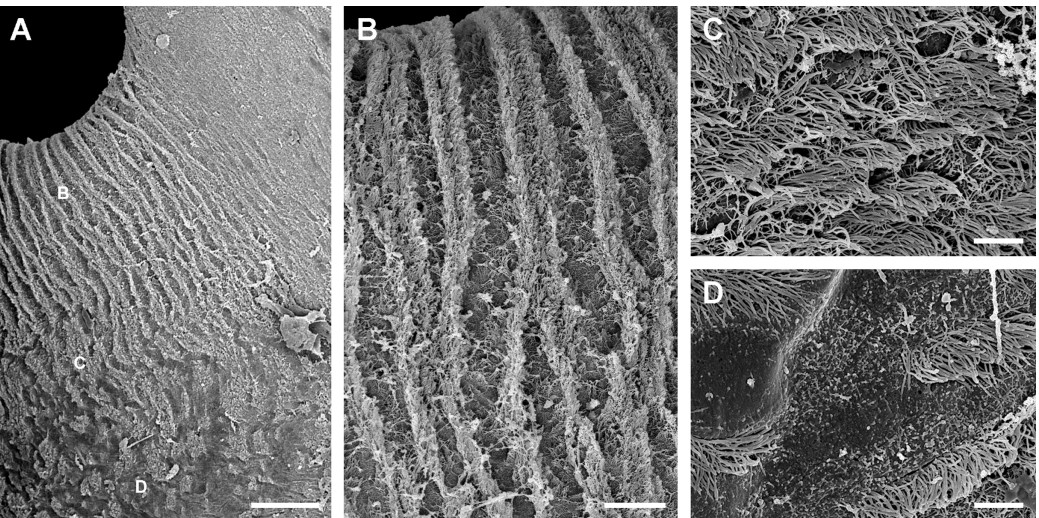

**Figure 5 Inner aspect of the pneumostome.** (A) Pneumostome border; the small letters B, C and D indicate the approximate regions shown in the corresponding panels at higher magnification. (B) Ciliary rows directed towards the pneumostome opening. (C) Ciliary patches in a region between those of the ciliary rows and of the pavement epithelium. (D) Pavement cells separated by rows of ciliary cells; this region abuts with the typical pavement epithelium of the respiratory lamina (Fig. 4B), whose cells are separated by grooves. Scanning electron microscopy. Scale bars represent: (A) 100 μm; (B) 25 μm; (C and D) 5 μm.

## The respiratory lamina: the gas-exchange barrier

A diagram overviewing our findings on the cellular components of the respiratory lamina and underlying structures is presented in Fig. 6. Pavement cells (Figs. 7A–7D), which correspond to those seen under SEM in Figs. 4A and 4B, cover the inner surface of the lung cavity. They show rather electron-dense and middle-sized nuclei, usually located near the intercellular grooves or in the grooves themselves. Both desmosome-like and septate junctions join the pavement cells, which emit numerous and short microvilli, particularly at or near the grooves (Figs. 7A and 7B). The cytoplasm of pavement cells has a low organelle content and extends as a thin layer overlying an anastomosing network of small blood spaces that rests over a fibromuscular layer made of muscle fibers in a collagen matrix. The resulting blood–gas barrier (80–150 nm thick) is made by the thin cytoplasmic layer and its corresponding lamina densa (Figs. 7A–7D). Also, irregular cytoplasmic extensions deepen at the grooves, sometimes reaching the fibromuscular layer, so that dome-shaped interconnected microsinuses are formed (Fig. S2). Hemocytes in the sinus blood show prominent pseudopodia and phagosomes (Fig. 7D), which seem involved in the phagocytosis of a particulate material of variable size and appearance that occurs in the sinus blood.

Muscle and collagen fibers appear sectioned both transversally and longitudinally in the fibromuscular layer (Figs. 7E and 7F). Neurite bundles with vesicles of moderate to high electron density and glial cells with granules of high electron density (Fig. 7F) occur in this fibromuscular layer, frequently in the proximity of the ciliary tufts, but no evidence of

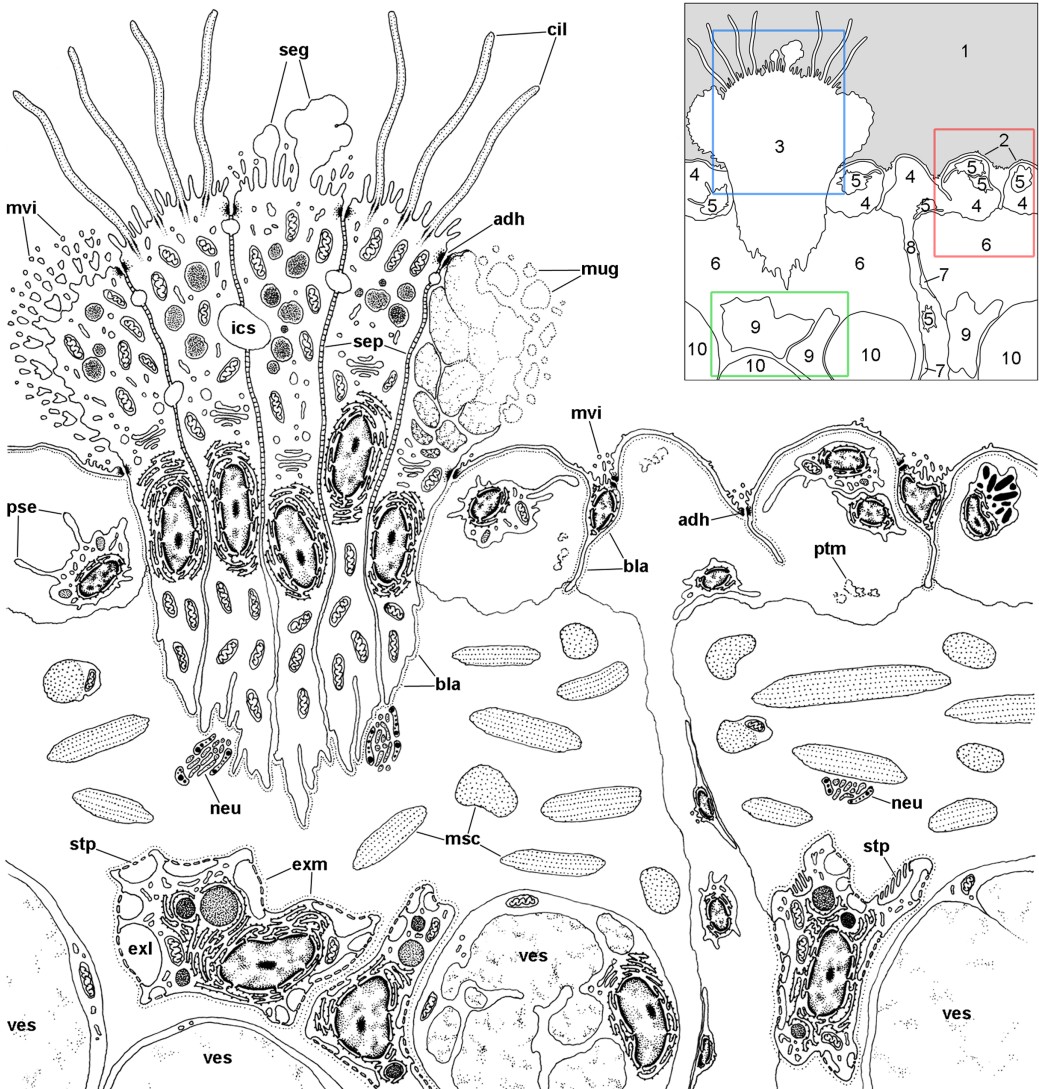

**Figure 6 Overview of the respiratory lamina and underlying tissues.** The diagram depicts a portion of the respiratory lamina including a ciliary tuft and the underlying tissues. The collagen matrix of the fibromuscular tissue is not shown for clarity. The main structures are indicated in the thumbnail in the right-upper corner. The boxes enclose the structures described in detail in Figs. 7 (red), 8 (blue), 9 and 10 (green). Abbreviations: 1, pulmonary cavity; 2, pavement cells; 3, ciliary tuft; 4, blood sinuses of the respiratory lamina; 5, hemocyte; 6, fibromuscular tissue; 7, endothelial-like cell; 8, radial sinus; 9, rhogocyte; 10, storage cell; adh, adherent junction; bla, basal lamina; cil, cilia; exl, extracellular lacunae; exm, extracellular matrix; ics, intercellular space; msc, muscle cells; mug, mucin granules; mvi, microvilli; neu, neurite bundle; pse, pseudopodia; ptm, particulate material; seg, secretory globules; sep, septate junctions; stp, slit apparatus; ves, vesicles of storage cells.

intraepithelial innervation (as that seen in the gill, *Rodriguez et al., 2019*) could be discerned in the lung.

It should be noted that this layer of fibromuscular tissue separates the respiratory lamina from the large blood sinuses that supply the lung. Radial sinuses originating from them traverse the fibromuscular layer conveying blood to the sinus of the respiratory lamina (Figs. 1B and 1C). These radial sinuses are of varying length, but they are much

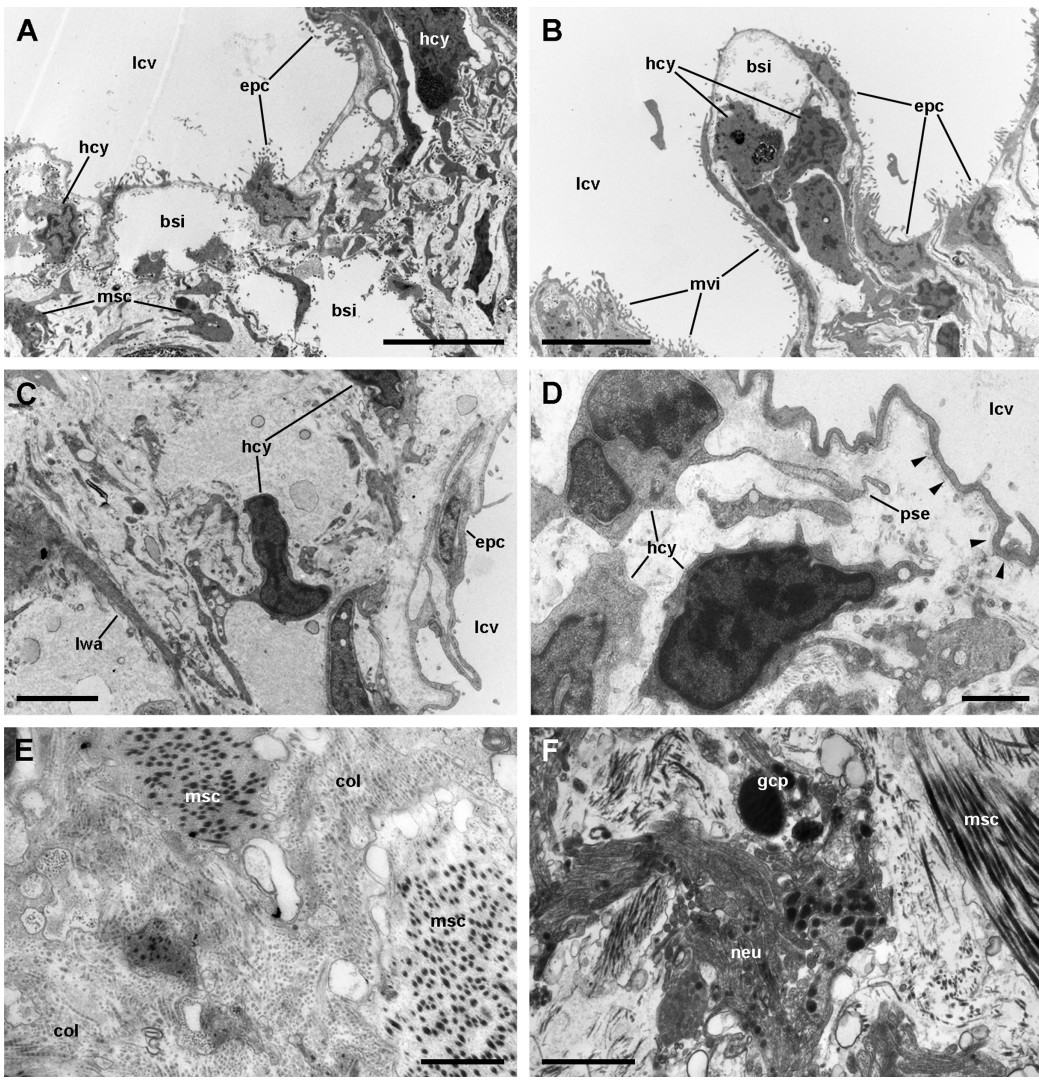

**Figure 7 The respiratory lamina.** (A–D) The pavement epithelium and the underlying sinus. The blood–gas barrier simply consists of an epithelial cell with its basal lamina (arrowheads in (D)) and thus it is very thin (~80–150 nm). The sinuses contain blood plasma and hemocytes and an abundant particulate material; the hemocytes exhibit pseudopodia and phagosomes and appear phagocytizing these particles. (E) Collagen fibers and intertwined muscle cells in the fibromuscular layer. (F) Neurites containing vesicles of moderate to high electron density often occur in the fibromuscular layer. Abbreviations: bsi, blood sinuses of the respiratory lamina; col, collagen matrix; epc, epithelial cells; gcp, glial cell processes; hcy, hemocytes; lcv, lung cavity; lwa, lateral wall of a blood sinus; msc, muscle cells; mvi, microvilli; neu, neurite bundle; pse, pseudopodium. Transmission electron microscopy. Scale bars represent: (A and B) 5 μm; (C) 2 μm; (D–F) 1 μm.

narrower than the main sinuses, and in certain occasions, there are direct connections between the main blood sinuses and the blood spaces of the respiratory lamina (Fig. 1A). We found sparse endothelial-like cells lining the radial sinuses (Fig. 6).

## The respiratory lamina: the ciliary tufts

Ciliated cells are the main cell type composing the tufts; they bear both long cilia and short microvilli (Fig. 8A). Secretory cells show granules with varying electron density (Fig. 8B).

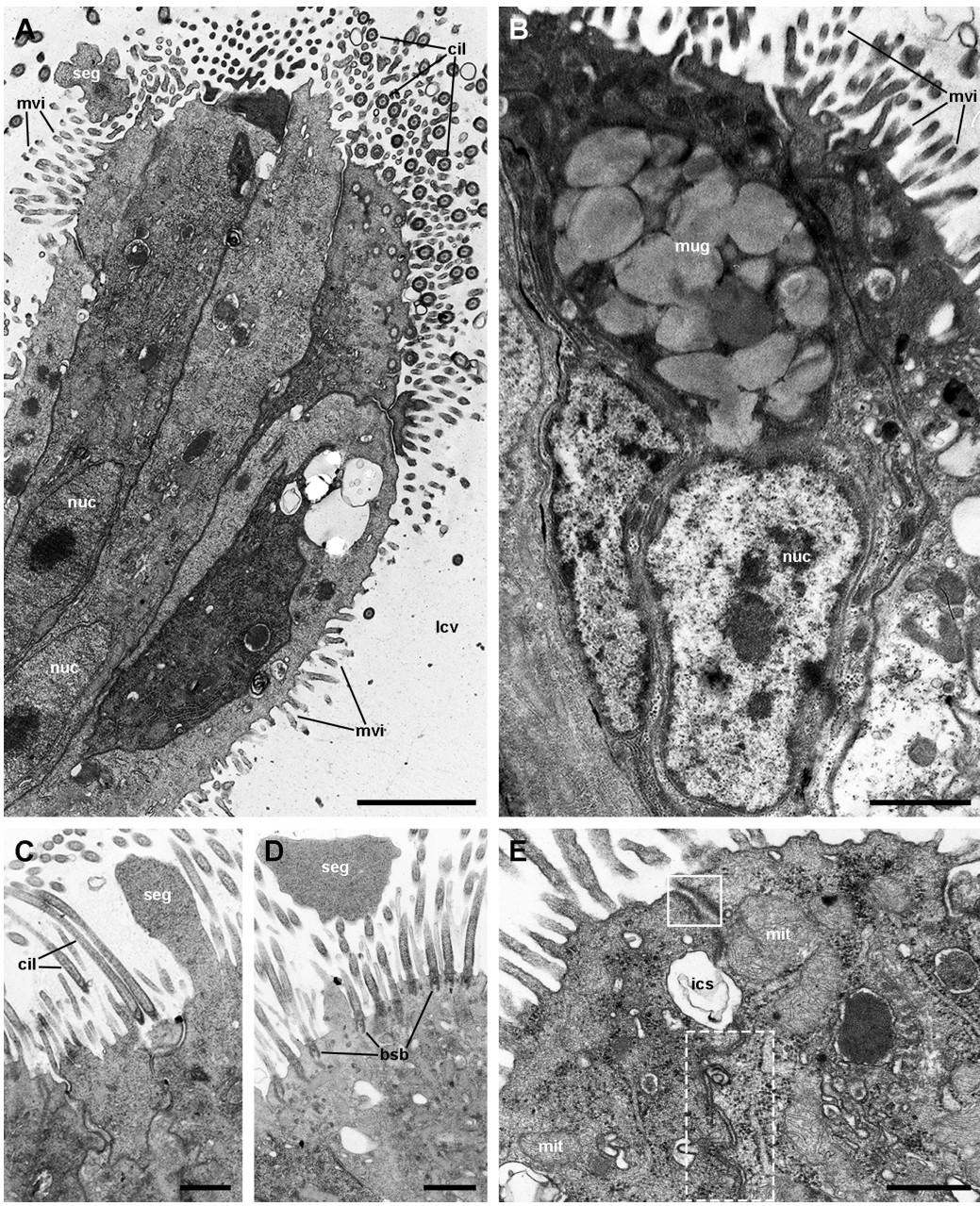

**Figure 8 Ciliary tufts of the respiratory lamina.** Aggregates of columnar cells, predominantly ciliary cells, protrude from the respiratory surface. Ciliary cells also show short, finger-like microvilli intercalated between the cilia (A; see also Fig. S2). Other cellular types are: (1) microvillar cells, which bear thick, often ramified, microvilli (A–D); (2) secretory granule cells (B; see also Fig. S2); and (3) secretory apocrine cells (C and D). Both desmosome-like ((E), solid box) and septate junctions ((E), dashed box) occur in the lateral domains. Nuclei in the cellular aggregates are rather uniform in shape. Abbreviations: bsb, basal bodies; cil, cilia; ics, intercellular space; lcv, lung cavity; mit, mitochondria; mug, mucin granules; mvi, microvilli; nuc, cell nucleus; seg, secretory globules. Transmission electron microscopy. Scale bars represent: (A and B) 2 μm; (C and D) 1 μm; (E) 500 nm.

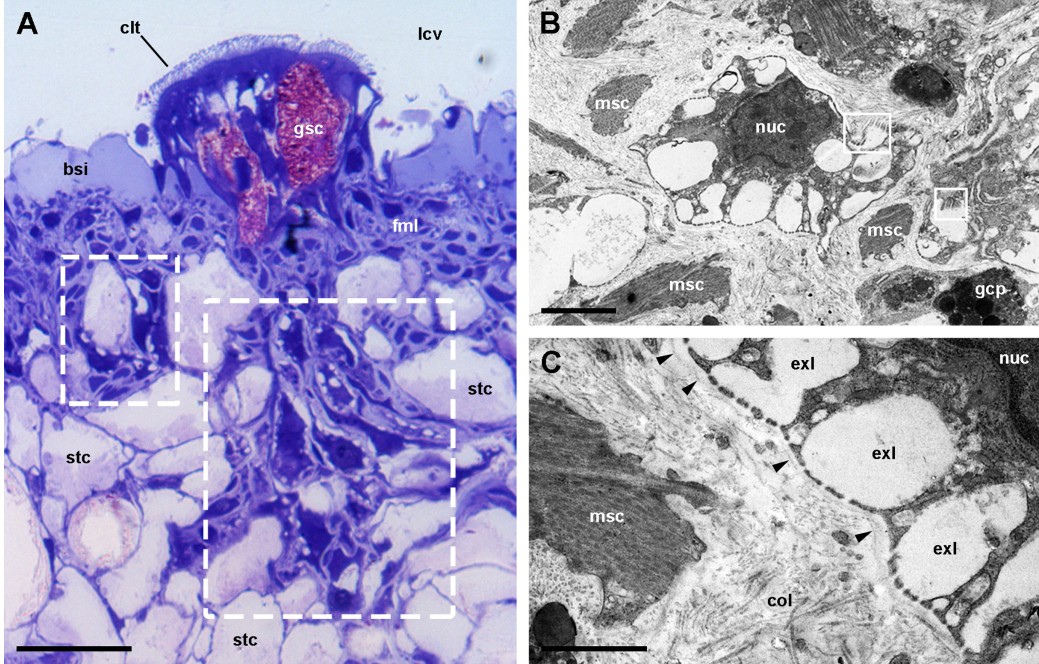

**Figure 9  Rhogocytes (I).** (A) Rhogocyte clusters at the base of a ciliary tuft (large dashed box) and associated to a storage cell (small dashed box). (B) A rhogocyte and the cytoplasmic extension of another one, showing the extracellular lacunae with the typical "slit apparatuses"; they are surrounded by a lamina densa and extracellular matrix where muscle fibers and storage cells are embedded. Most components of the slit apparatus (i.e., bars and slits) appear sectioned transversally, but the two dashed boxes show some longitudinally sectioned. (C) Detail of the extracellular lacunae of the same rhogocyte. Arrowheads indicate the extracellular matrix that surrounds the rhogocyte. (A) Toluidine blue stain and (B and C) Transmission electron microscopy. Abbreviations: bsi, blood sinus of the respiratory lamina ; clt, cellular tuft; col, collagen matrix; exl, extracellular lacunae; fml, fibromuscular layer; gcp, glial cell process; gsc, granular secretory cell; lcv, lung cavity; msc, muscle cell; nuc, nucleus; stc, storage cell. (A) Toluidine blue stain and (B and C) Transmission electron microscopy. Scale bars represent: (A) 20 μm; (B) 2 μm; (C) 1 μm.                         

Some of these secretory cells show metachromatic clumps when stained with toluidine blue (presumably mucin, Fig. S2). Another type of secretory cells shows thick, globule-forming protrusions of the apical cytoplasm that detach to the pulmonary cavity (Figs. 8C and 8D; Fig. S2B), that is, as a kind of apocrine secretion. Under SEM (Fig. 4D), these globules are seen merging into larger masses. There are also microvillar cells in the tufts, which show rather long and ramified microvilli (Figs. 8A and 8B), and whose cytoplasm contains only some mitochondria, which are more abundant in the subnuclear region. These cells resemble the β-cells found in the gill epithelium of *P. canaliculata* (*Rodriguez et al., 2019*), although the cytoplasm of those is packed with mitochondria. Cell-to-cell contacts comprise both desmosome-like and septate junctions, which keep the tuft cells together, between them and with the pavement cells of the respiratory lamina.

## Below the respiratory lamina: rhogocytes and storage tissue

We found rhogocytes either isolated or grouped, near the roots of the ciliary tufts and associated with radial sinuses and cells of the storage tissue (Fig. 9A) (*Giraud-Billoud et al., 2008*). Rhogocytes varied in size and shape and exhibited the characteristic extracellular

lacunae enclosed by cytoplasmic bars separated by slits (*Haszprunar, 1996*; *Kokkinopoulou et al., 2014*) (Figs. 9B, 9C and 10). Rhogocytes were enclosed within a lamina of the extracellular matrix (Fig. 9C, arrowheads) and often surrounded by dense bundles of collagen fibers (Figs. 10A–10C). They also showed long cytoplasmic extensions that frequently run parallel to each other (Fig. 10B). Rhogocytes' nuclei were large, with heterochromatic clumps and nucleoli (Figs. 9B, 10B, 10C and 10E); the cytoplasm showed a remarkable arrangement of both smooth and rough endoplasmic reticula (SER and RER; Fig. 10). The SER was made of rather wide tubuli containing a flocculent material and that communicated with the extracellular lacunae. Notably, the slits of these lacunae do not open directly to the surrounding extracellular fluid or the hemocoel, because rhogocytes and their cytoplasmic projections are encased by an external lamina or are in contact with dense collagen tissue (Figs. 9C and 10). The content of the RER tubuli was of an electron density similar to that of the surrounding cytoplasm (Fig. 10A). Wide cisternae formed in the course of both SER (Fig. 10E) and RER tubuli (Fig. 10B). Groups of glycogen clumps (Fig. 10A), as well as some well-delimited cytoplasmic globules, with varying sizes and electron densities (Figs. 10A, 10C and 10D), also occurred in the rhogocytes of the examined *P. canaliculata* individuals.

A summary of the lung structures in *P. canaliculata* and their proposed functions is presented in Table 1.

## DISCUSSION

### The lungs of the Ampullariidae in the context of the Gastropoda

Cavities regarded as lungs occur in the major gastropod clades Neritimorpha, Caenogastropoda and Heterobranchia (*Ponder, Lindberg & Ponder, 2019*). The majority of lung-bearing gastropods, however, are comprised within the Stylommatophora (superorder Eupulmonata) and the "basommatophora" (superorder Hygrophila) (*Bouchet et al., 2017*), in which the pallial cavity has acquired the role of an air-breathing organ and is partly closed by extensions of the mantle edge. In these cases, the lung and the pallial cavity are homologous structures (*Ruthensteiner, 1997*) and the pneumostome opens to the exterior (Fig. 11A).

In the particular case of the Ampullariidae, however, the lung is a flattened sac exhibiting distinct roof and floor regions, and a pneumostome that opens through the floor, connecting the lung cavity with the mantle cavity (Fig. 11B) (*Hylton Scott, 1957*). Thus, this pneumostome is not homologous to the pneumostome of other lung-bearing gastropods. Indeed, the ampullariid lung arises during development as an invagination of the mantle cavity roof, between the primordia of the gill and the osphradium, at least in the genera *Pomacea* (*Brooks & McGlone, 1908*; *Koch, Winik & Castro-Vazquez, 2009*), *Marisa* (*Demian & Yousif, 1973*) and *Pila* (*Ranjah, 1942*). Hence, the embryonic inner mantle epithelium lines the lung floor on both sides, but only the inner side differentiates as a "respiratory lamina" (sensu *Rodriguez et al., 2018*) (see Figs. 11A and 11B for comparison).

To our knowledge, and before the current study, the detailed histology of these presumptive gas exchanging areas had being adequately studied in just one tropical slug

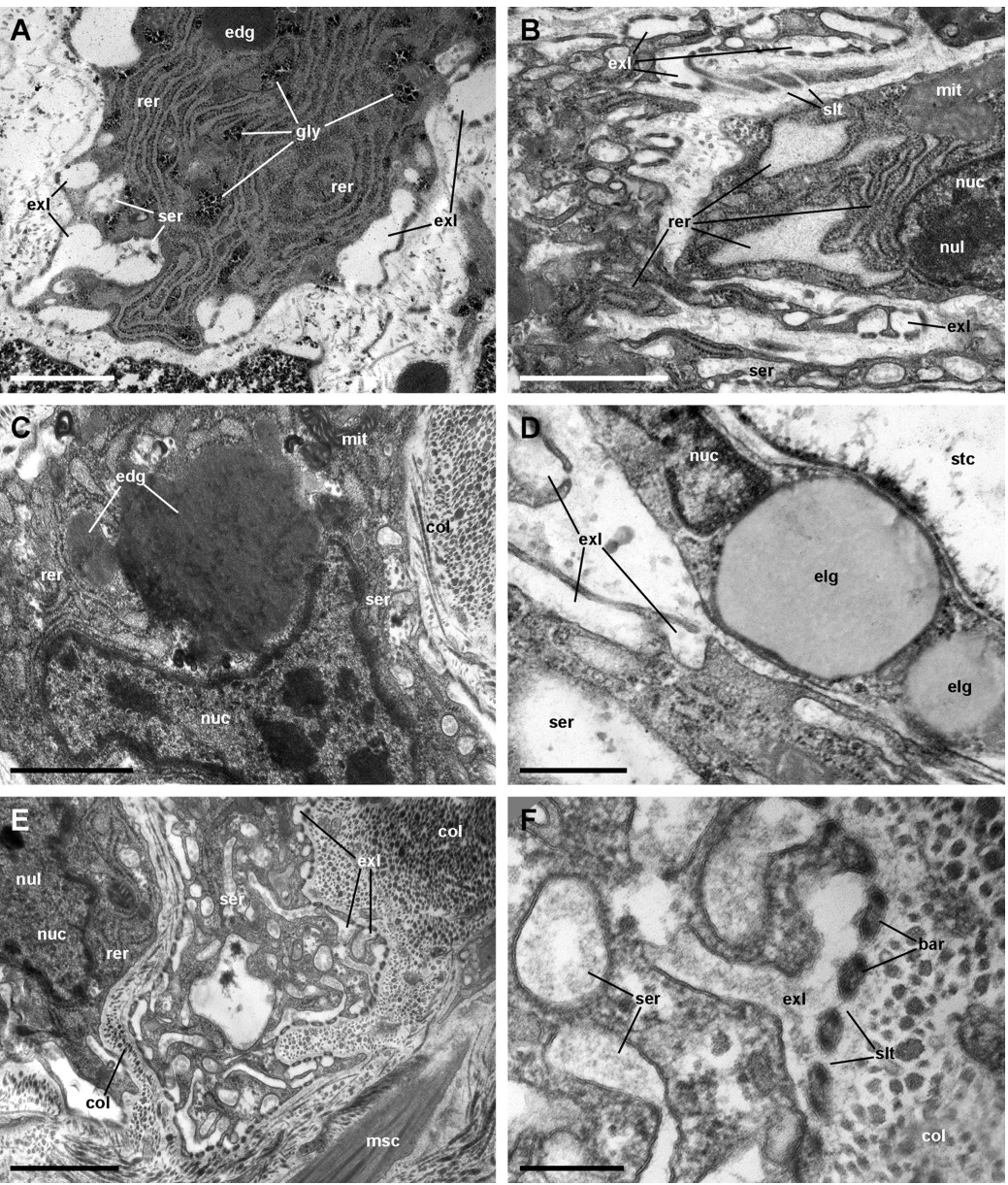

**Figure 10 Rhogocytes (II).** (A) Rough endoplasmic reticulum with interspersed glycogen clumps; profiles of the smooth endoplasmic reticulum are also seen in the vicinity of the extracellular lacunae. (B) Wide cisternae of the rough endoplasmic reticulum, and a large nucleolus within a rounded nucleus; cytoplasmic extensions found in the vicinity (probably from other rhogocytes) contain rough endoplasmic reticulum and extracellular lacunae. (C) Two electron-dense globules appear merging near an irregularly shaped nucleus, with heavy heterochromatic clumps. (D) Cytoplasmic extensions close to a storage cell showing a wide cistern of the smooth endoplasmic reticulum, extracellular lacunae and two electron-lucent globules. (E) A cytoplasmic region showing a wide cistern of the smooth endoplasmic reticulum as wells as multiple tubuli, some of which connect with extracellular lacunae. (F) Close up of a connection of the smooth endoplasmic reticulum with an extracellular lacuna; the slit apparatus opens to a region of tightly packed collagen fibers. Abbreviations: bar, bar of the slit apparatus; col, collagen matrix; edg, electron-dense globule; elg, electron-lucent globules; exl, extracellular lacunae; gly, glycogen; mit, mitochondria; msc, muscle cell; nuc, nucleus; nul, nucleolus; rer, rough endoplasmic reticulum; ser, smooth endoplasmic reticulum; slt, slit of the slit apparatus; stc, storage cell. Transmission electron microscopy. Scale bars represent: (A–C) and (E) 1 μm; (D) 500 nm; (F) 200 nm.

**Table 1  Lung structures and their proposed functions.**

| Structure | | Proposed function |
|---|---|---|
| **Respiratory lamina** | | |
| Pavement epithelium | Pavement cells | Form the blood-gas barrier (80–150 nm thick) |
| Ciliary tufts | Ciliated cells | Lubrication and cleansing of the respiratory lamina |
| | Microvillar cells | Anchoring of the pavement epithelium to the fibromuscular layer |
| | Secretory orthochromatic cells | |
| | Secretory metachromatic cells | |
| Blood sinus | — | Blood flow allowing gas exchange |
| | | Facilitation of hemocyte-antigen contact and the consequent immune response |
| Fibromuscular layer | Muscle cells | Supporting portion (or "hard part", sensu *Maina & West, 2005*) of the respiratory lamina |
| | Fibroblasts | |
| | Collagen matrix | |
| | Neurites | |
| | Glial cells | |
| **Underlying tissues** | | |
| Perivascular tissue | Storage cells | Storage of glycogen (*Andrews, 1965*) and uric acid (*Giraud-Billoud et al., 2008*) |
| | Rhogocytes | Proposed role in metal detoxification and hemocyanin synthesis in other gastropods (*Haszprunar, 1996*) |
| Blood sinuses and veins | — | Blood transport through the lung, to and from the respiratory lamina |
| Muscular layer (single in the lung roof and double in the lung floor) | Muscle cells | Lung volume adjustment |
| | Nerves | |
| | Collagen matrix | |

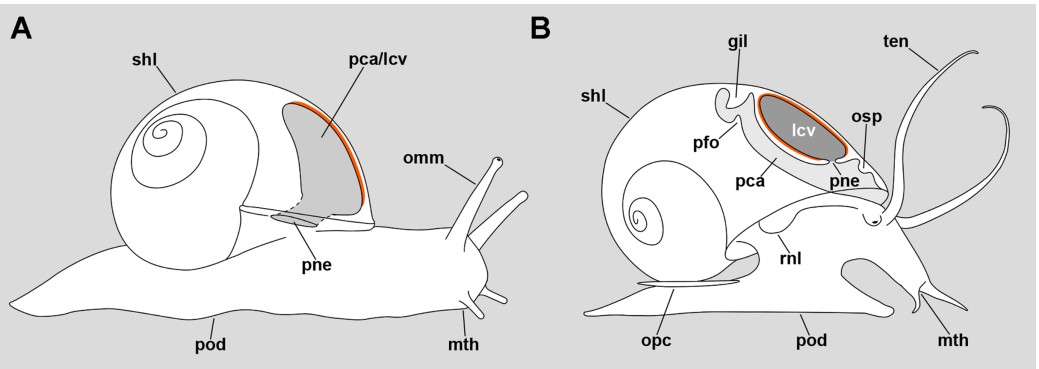

**Figure 11 Comparative diagrams of the lungs of an helicid and an ampullariid.** (A) Outline of an individual *Helix pomatia* (Heterobranchia: Helicidae) showing a section of its pallial/lung cavity. (B) Outline of an individual *P. canaliculata* with a section through its pallial and lung cavities. The location of the respiratory epithelium, in each case, is shown in orange. The representation of structures has been simplified for clarity. Abbreviations: gil, gill; lcv, lung cavity; mth, mouth; omm, ommatophore; opc, operculum; osp, osphradium; pca, pallial cavity; pfo, pallial fold; pne, pneumostome; pod, foot; rnl, right nuchal lobe; shl, shell; ten, tentacle.

(*Maina, 1989*; *Polytoxon robustum*, Urocyclidae, as *Trichotoxon copleyi*, a junior synonym) and one ampullariid snail (*Lutfy & Demian, 1965*; *Marisa cornuarietis*). The study in *P. robustum* was made using SEM and TEM, and the inner surface of the lung

(the presumptive gas exchanger) appears covered by approximately parallel ridges, which appear hollow in TEM sections and that *Maina (1989)* called "capillaries". They are lined externally by pavement cells, much similar to those lining the blood respiratory sinuses in *P. canaliculata*. Also, the blood sinuses contain a particulate material among the hemocytes, similar to that seen in *P. canaliculata*. We interpret they are cell debris that the slow blood flow accumulates in the sinuses.

Instead, the study in *M. cornuarietis* (*Lutfy & Demian, 1965*) was made exclusively using light microscopy of paraffin sections, which we think adds merit to that outstanding paper, in which they accurately described the blood sinuses of the respiratory lamina but, unfortunately, they mistakenly equated the storage cells of the lung wall with blood lacunae. In fact, the cytoplasmic contents of these large cells may be extracted during histological processing, and the remaining cytoplasmic rim may give the false appearance of the wall of blood sinuses. Another important limitation of that paper is its strictly descriptive character and, consequently, the fact that the authors did not advance any functional interpretation. In both studied Ampullariidae (*Lutfy & Demian, 1965*; and this article), the vascularized area involves all the inner surface of the lung cavity.

## The lung floor, the gill, and the adaptation of *Pomacea canaliculata* to air breathing

The lung roof and the lung floor receive blood from partly different origins. The lung roof receives blood from right afferents (Fig. 2E), while the lung floor receives blood from right afferents that have already traversed the gill and from left afferents that come directly (Fig. 2F). Deoxygenated blood from the visceral hump reaches the *afferent branchial vein* through the *right pallial* and *rectal sinuses*. That vein not only supplies the gill but, after leaving it, this vein is termed the *afferent pulmonary vein* (*Andrews, 1965*) and distributes blood to all of the lung roof and to the anterior and right side of the lung floor. Deoxygenated blood conveyed by blood sinuses from the mantle edge also reaches the lung roof through the *afferent pulmonary vein*. After oxygenation in the respiratory lamina of the lung roof, the blood is drained with that coming from the gill through a common efferent route, that is, the *efferent pulmobranchial vein* (Fig. 2A). Instead, the lung floor receives oxygenated blood from the gill leaflets through the *ventral afferent pulmonary vein* (Fig. 2B, right sinuses). It also receives deoxygenated blood from the left side of the body, through several slender *left sinuses* and from a large sinus that collects blood from the siphon and the left mantle border (Fig. 2B, left sinuses).

When the animal is submerged and the gill leaflets are deployed, oxygenated blood may reach the auricle directly through the *efferent pulmobranchial vein*. However, when the animal is outside of the water, the gill leaflets collapse and stick together, thus drastically reducing both the circulation through the leaflet sinuses and the available surface for gas exchange. Then, the blood would shunt through the *basal leaflet sinuses* (*Rodriguez et al., 2019*) converging into the *ventral afferent pulmonary vein*, which would distribute blood in the right side of the lung floor. Also, even when breathing in water, blood conveyed by the *rectal* and *right pallial sinuses* may traverse the gill leaflets, also reaching the lung floor through the *ventral afferent pulmonary vein*.

A possible mechanism in the evolution towards water independence, might have been the development of the *ventral afferent pulmonary vein* (see Figs. 1A and 2C) which, as we have commented above, would allow the shunt of blood from the gill towards the lung floor, and that it may have derived from the longitudinal division of the *efferent pulmobranchial vein* (*Andrews, 1965*). Also, *Simone (2004)* found the longitudinal splitting of this vein in *Pomacea lineata* and *Pomacea sordida*. Other ampullariid species this author investigated with negative results were *Pomacea crosseana*, *Pomacea curumim*, *Pomacea megastoma* and *Pomacea scalaris*, as well as *Marisa planogyra* and *Felipponea neritiniformis* (*Simone, 2004*). Also, *Demian (1965)* obtained negative results for *M. cornuarietis*. Admittedly, this number of species is far from representative of this highly diverse family. And also, it should be considered that only *P. canaliculata* (*Seuffert & Martín, 2009*; *Seuffert & Martín, 2010*) and, to some extent, *P. lineata* (*Little, 1968*) have been studied ecophysiologically. Thus, further investigations in a much larger number of species, from both morphological and functional viewpoints, would be needed to test this hypothesis.

## The respiratory lamina: how it can be both thin and strong

In general, a blood–gas barrier must have both a thin and an extensive surface to facilitate gas diffusion, but it should be sturdy enough to withstand the rough work of respiratory movements and currents. Comparative studies have stressed that this is achieved by structures that combine a "soft" and a "hard part", the former to allow gas-exchange, and the latter to keep the soft part in place, and in functional conditions (*Maina & West, 2005*).

In *P. canaliculata*, the thin gas exchange membrane (80–150 nm width, Fig. 7D) needs to be anchored to a "hard part" (*Maina & West, 2005*), which in this case is the fibromuscular layer (Figs. 7A–7D). This is achieved by the intercellular junctions that bind the pavement cells to the epithelial cells of the ciliary tufts, which are themselves deeply rooted in the fibromuscular layer at frequent intervals (Fig. 4A), and thereby may be acting as rivets that keep the epithelial and connective parts together (Fig. 4A). We have also occasionally observed some fibroblast cells in direct contact with the basal part of pavement cells, and that may be an additional structural component that needs to be explored further.

However, other important functions of the ciliary tufts should be those related to their secretions and to the ability of their cilia to spread them over the surface of pavement cells. Possible roles would be the agglutination of dust particles that have entered with air, and that would be expelled by the ciliary rows of the pneumostome (Fig. 5), as well as the lubrication of the delicate respiratory lamina during breathing movements. Besides that, a phospholipidic surfactant, found in the lungs or tracheoles of a wide variety of animals (*Orgeig et al., 2007*), has been reported in the lung of the garden snail *Cornu aspersum* (Stylommatophora, Helicidae) (*Daniels et al., 1999*). This should encourage the search for compounds with a similar function in the ciliary tufts' secretions in *P. canaliculata*.

Taken together, our histological and ultrastructural findings on the respiratory organs may constitute the morphological basis to explain why *P. canaliculata* is an obligate air-breather.

## The respiratory lamina in relation to hemocyte nodulation, rhogocytes and storage tissue

The particulate material occurring in the sinuses of the respiratory lamina is likely composed of cell debris collected when blood traverses the body vascular network. Crossing these wide spaces before reaching the lung should cause the blood to flow at low pressure and speed in the respiratory lamina. That, along with the intricacy of this structure, may promote the contact of circulating phagocytes (*Cueto et al., 2015*) with blood-borne microorganisms and foreign particles. In turn, this may initiate the defensive reaction of hemocyte nodulation we have shown elsewhere (*Rodriguez et al., 2018*). Also, both hematopoietic stem cells and proliferating cells are known to occur in the circulation of *P. canaliculata* (*Rodriguez et al., 2020*) and these cells may be attracted, or even trapped in these narrow spaces, where nodulation may start. In this way, both the kidney islets and the lung's respiratory lamina may act as immune barriers and hematopoietic sites (*Rodriguez et al., 2018*).

Rhogocytes occurred near the roots of the ciliary tufts and in the storage tissue underneath the respiratory lamina. Storage cells have been proposed to accumulate glycogen (*Andrews, 1965*) and urates (*Giraud-Billoud et al., 2008*; *Vega et al., 2007*) in *P. canaliculata* or, more recently, calcium carbonate in *P. maculata* (*Mueck, Deaton & Lee, 2020a*). Instead, rhogocytes have been implicated in metal detoxification and hemocyanin synthesis (*Haszprunar, 1996*; *Kokkinopoulou et al., 2015*). Like rhogocytes, storage cells have received several names in the past, and at least one of them was shared (vesicular cells, =*Blasenzellen*), suggesting they have been sometimes considered the same under the light microscope. However, they are clearly different under the electron microscope, and the existence of the "slit-apparatus", which delimits the extracellular lacunae, is distinctive for rhogocytes (*Kokkinopoulou et al., 2015*). In *P. canaliculata*, *Giraud-Billoud et al. (2008)* have described the location of storage cells in several organs and tissues, as well as their association with high tissue levels of uric acid. *Vega et al. (2007)* were able to isolate crystalloids from perivascular storage tissue of the digestive gland and found they were mainly composed of urates and protein. The close association of rhogocytes with storage cells has been intriguing to us, because we have occasionally observed rhogocytes in other tissues too, such as the aortic ampulla and the perivascular tissue of the digestive gland, and all of these organs also contain storage cells (*Giraud-Billoud et al., 2008*). In *P. canaliculata*, there is substantial evidence for a role of the accumulated uric acid in counteracting the oxidative burst that accompanies the arousal from estivation and hibernation (*Giraud-Billoud et al., 2011*; *Giraud-Billoud et al., 2018*; *Giraud-Billoud et al., 2013*), while *Mueck, Deaton & Lee (2020a)* have proposed a role of the calcareous crystalloids in buffering the lowering of pH that occurs during estivation in *P. maculata*.

Regarding the possible role of rhogocytes in hemocyanin synthesis, we must first say we could not discern any images of hemocyanin polymers in lung rhogocytes or in their vicinity, like those described in other gastropods (*Albrecht et al., 2001*; *Kokkinopoulou et al., 2015*). Instead, in a preceding publication (*Rodriguez et al., 2018*), we did observe hemocyanin polymers replenishing the lacunae of the renal hemocyte islets, even though we observed no rhogocytes in renal tissue. In the current study, a flocculent material was contained in SER vesicles and tubuli and was apparently secreted into the extracellular lacunae (Figs. 10E and 10F), and one may wonder whether this material could correspond to hemocyanin monomers or oligomers to be transported in blood to the renal hemocyte islets, where they would be polymerized. However, further research is needed to elucidate the exact pathways for the synthesis, polymerization, storage and bioavailability of hemocyanin.

## The lung of *P. canaliculata* in the evolutionary framework of the Ampullariidae

Our knowledge of the phylogenesis of the Ampullariidae is based on morphological (*Berthold, 1991*), behavioral (*Hayes et al., 2009*) and molecular studies (*Hayes, Cowie & Thiengo, 2009*) that have disclosed much of the intergeneric relationships within the family. However, an all-encompassing, comparative study of the respiratory organs in Ampullariidae is still lacking. That said, some general trends might be identified with the available data within the framework of the intergeneric relationships.

The lungs of the basal, Old World ampullariid genera *Afropomus* and *Saulea* are less elaborate than those of the derived, New World genera *Pomacea* and *Marisa* (*Berthold, 1988*, *1991*). This evolutionary trend in lung morphology may parallel the evolution of the left nuchal lobe into an extensible siphon, and of the corresponding siphonal mode of aerial respiration, which has not been reported for the basal genera. It also correlates with the oviposition behavior and the characteristics of the egg clutches in different genera (*Hayes et al., 2009*), as well as with the occurrence of other adaptations in the eggs related to oviposition outside water, such as a calcareous capsule and defensive perivitelline proteins (*Brola et al., 2020*; *Dreon et al., 2013*; *Hayes et al., 2015*; *Sun et al., 2019*).

Another aspect that is worth noting is the complexity of the vascular network. "Highly vascularized lungs" have been attributed by authors to species of the derived genera, though this expression lacks the needed precision. For instance, species showing large sinuses traversing the lung floor may impress the observer as being "highly vascularized", but what really matters for respiration is the organization of the respiratory lamina. Notably, however, such structure has only been studied to some extent in *M. cornuarietis* (*Lutfy & Demian, 1965*) and, more in depth, in *P. canaliculata* (*Rodriguez et al., 2018*; and this article).

The varying thickness of the lung floor of the Ampullaridae, with a tendency to get thicker in derived species, was noted by some authors (*Berthold, 1991*; *Simone, 2004*); however, more data are needed to make meaningful comparisons. In fact, the main tissues affecting the thickness of the lung floor are (1) the thick muscle layer that underlies the mantle cavity epithelium in front of the pneumostome (see Fig. 1A), and (2) the layer of

storage cells (see Figs. 1B and 1C). Only the first one may have a functional role in respiration, because its contraction or relaxation may affect the volume of the lung cavity. The second one has, at least in *P. canaliculata*, a role in the antioxidant protection during arousal from estivation (*Giraud-Billoud et al., 2011*; *Giraud-Billoud et al., 2013*) and hibernation (*Giraud-Billoud et al., 2018*).

Regarding the pneumostome, the ancestral condition among the Ampullariidae may be represented by a round, broad and thus presumably non-closable structure, placed centrally and slightly to the right of the lung floor, as it is represented in the basal genus *Afropomus* (*Berthold, 1988*). Instead, the derived condition may be characterized by a slit-like, small, and closable pneumostome, placed anteriorly and to the left, near the siphon base, as it occurs in the genera *Marisa* and *Pomacea* (*Berthold, 1991*). It is worth noting that in *Saulea*, the pneumostome is rounded but small and controlled by a muscular ring, and therefore it may represent an intermediate condition (*Berthold, 1991*).

The lungs of *Pomacea* species are likely among the largest and most elaborate within the family Ampullariidae. In *P. canaliculata*, the lung occupies most of the roof of the pallial cavity (Fig. 1A) and has a respiratory lamina with a prominent network of supplying and draining vessels (Figs. 1A and 2). The pneumostome is small, slit-like, and can be closed by overlapping its two lips. It is muscular and is innervated by two nerve branches from the *supraesophageal ganglion* (Fig. 3). As already mentioned, the musculature at the left anterior part of the lung floor is thicker than in the rest of the lung (Fig. 1A), and may be engaged in bringing the pneumostome forward (*Andrews, 1965*), close to the siphon base during the siphonal mode of respiration. The abundance of cilia near the pneumostome and its orderly arrangement suggest they serve the function of expelling the small particles that enter with air and are agglutinated by mucus from the ciliary tufts.

The embryonic development of the lung has been studied in representatives of three of the major clades of Ampullariidae, namely the *Pila* clade (Asian), and the *Marisa* and *Pomacea* clades (South American) (*Hayes, Cowie & Thiengo, 2009*) and this may also help to put our findings in an evolutionary framework. The lung develops in *Pila globosa*, *P. canaliculata* and *M. cornuarietis* as an invagination of the dorsal wall of the mantle cavity (*Brooks & McGlone, 1908*; *Demian & Yousif, 1973*; *Koch, Winik & Castro-Vazquez, 2009*; *Ranjah, 1942*). However, while the mantle cavity in *Pila globosa* and *P. canaliculata* originates at the posterior end of the embryo, long before torsion begins, it originates in *M. cornuarietis* after the onset of torsion as a depression on the right dorsolateral side of the embryo (*Demian & Yousif, 1973*). Other marked differences in the developmental timing of the respiratory organs are brought out by the findings of *Demian & Yousif (1973)* and *Koch, Winik & Castro-Vazquez (2009)* who found the lung develops after the gill and the osphradium in *P. canaliculata* and *M. cornuarietis*, in the initially narrow space between those organs, while *Brooks & McGlone (1908)* and *Ranjah (1942)* reported the lung, gill, and osphradium develop simultaneously in *P. paludosa* and *Pila globosa*. A greater asynchronicity occurs in the development of the lung in *Asolene platae*, which according to *Tiecher, Burela & Martín (2013)*, was not detectable in newly hatched juveniles and only became functional two months after hatching.

### Nerves controlling branchial and pulmonary respiration in *P. canaliculata*

The lung and the gill share innervation from both the *supraesophageal* and the *accessory visceral ganglia*. The former ganglion gives off branches that innervate the lung roof, the pneumostome, the osphradium, and each of the gill leaflets through the *leaflet nerves* (Fig. 3). The *accessory visceral ganglion* gives off several branches that innervate the lung floor and a single branch that runs all the way through the branchial base, namely the *branchial base nerve* (Fig. 3C).

The shared innervation as well as the interconnections between the *supraesophageal* and the *accessory visceral ganglia* might also provide the neural mechanisms for the behavioral shifts between branchial and pulmonary breathing, which is coupled with the deployment of the siphon for air breathing (*Bavay, 1873*, *1875*; *Bouvier, 1888*, *1891*; *Fischer & Bouvier, 1890*; *Jourdain, 1879*; *Sabatier, 1879*). Also, there may be local responses in the gill and lung. Responses in the gill may involve regulation of blood flow by the contraction of muscular trabeculae controlled by subepithelial innervation, and the modulation of the ionic/osmotic regulation function of the gill epithelium by intraepithelial innervation (*Rodriguez et al., 2019*). Such intraepithelial innervation does not occur in the lung's respiratory lamina. Indeed, neurite bundles and glial cell processes are only found near the roots of the ciliary tufts and in the fibromuscular layer that underlies the blood sinuses of the respiratory lamina (Figs. 7E and 7F). Those neurite bundles show vesicles of moderate to high electron density and are associated with glial cells bearing larger granules like those observed in the gill leaflets of *P. canaliculata* (*Rodriguez et al., 2019*). The responses in the lung would involve the control of secretory cells in the ciliary tufts, the contraction of the lung floor, or the closure of the pneumostome. These morphological studies may pave the way to the needed experimental physiological studies.

## CONCLUSIONS

1. The lung cavity is lined by the respiratory lamina, which is composed of a pavement epithelium enclosing a mesh of interconnected blood spaces, the floor of which is made by fibromuscular tissue. Cellular tufts composed of columnar ciliated, microvillar, and secretory epithelial cells are interspersed in the respiratory lamina.

2. The blood-gas barrier is 80–150 nm thick and it simply made by the epithelial cell and its basal lamina. The epithelial cells' extensions that form the barrier are almost devoid of organelles. This contrasts with the gill epithelium of this species, which is about 200-fold thicker and is characterized by mitochondria-rich cells and other features that are typical of transporting epithelia.

3. The respiratory lamina fulfils the requirements of an effective gas exchanger and this suggests the lung is the main site for oxygen uptake in *P. canaliculata*. Our present findings in the lung, together with the previous ones in the gill, may provide the morphological grounds to understand the obligate aerial respiration in this species.

4. In general, the blood supply to the lung floor would become more important when the passage through the gill leaflets is impeded, and blood shunts to the *ventral afferent pulmonary vein*. This might be a crucial role for this vein, which is known to occur in only three *Pomacea* species and may be related to the evolution of amphibiousness in the genus.

5. The lung and gill share a common innervation from the interconnected *supraesophageal ganglion* (lung floor and gill leaflets) and the *accessory visceral ganglion* (lung roof, pneumostome, and gill base). The shared innervation may provide the underlying morphofunctional mechanism for the shift between branchial and lung breathing in *P. canaliculata*.

6. We described rhogocytes, for the first time in an ampullariid, and discussed their association with storage tissue in other organs. Roles in heavy metals detoxification and hemocyanin production have been proposed for these cells.

## ACKNOWLEDGEMENTS

The authors appreciate the generous technical support from Sergio A. Carminati and the people at the Microscopy Facilities of IHEM-CONICET and MEByM-CONICET (Elisa M. Bocanegra, Norberto F. Domizio, Mabel Fóscolo, María Silvina Lassa and María Paula López).

### Funding

This work was funded by Universidad Nacional de Cuyo, M068 (to Alfredo Castro-Vazquez), and M086 (to Cristian Rodriguez). The funders had no role in study design, data collection and analysis, decision to publish, or preparation of the manuscript.

### Grant Disclosures

The following grant information was disclosed by the authors:
Universidad Nacional de Cuyo: M068 and M086.

### Competing Interests

The authors declare that they have no competing interests.

### Author Contributions

- Cristian Rodriguez conceived and designed the experiments, performed the experiments, analyzed the data, prepared figures and/or tables, authored or reviewed drafts of the paper, and approved the final draft.
- Guido I. Prieto conceived and designed the experiments, performed the experiments, analyzed the data, prepared figures and/or tables, authored or reviewed drafts of the paper, and approved the final draft.
- Israel A. Vega conceived and designed the experiments, analyzed the data, authored or reviewed drafts of the paper, and approved the final draft.

- Alfredo Castro-Vazquez conceived and designed the experiments, analyzed the data, prepared figures and/or tables, authored or reviewed drafts of the paper, and approved the final draft.

## Ethics

The following information was supplied relating to ethical approvals (i.e., approving body and any reference numbers):

The Institutional Committee for the Care and Use of Laboratory Animals (Comité Institucional para el Cuidado y Uso de Animales de Laboratorio, Facultad de Ciencias Médicas, Universidad Nacional de Cuyo) approved the procedures for snail culture, sacrifice, and tissue sampling (approval protocol N° 55/2015).

## Data Availability

Raw data is available at Figshare:

Rodriguez, Cristian; Prieto, Guido; Vega, Israel; Castro-Vazquez, Alfredo (2021): Lung structures of *Pomacea canaliculata*. figshare. Figure. DOI 10.6084/m9.figshare.12562550.v1.

A 3D model of the lung vasculature and innervation is available at Morphosource: DOI 10.17602/M2/M122573.

## Supplemental Information

Supplemental information for this article can be found online at http://dx.doi.org/10.7717/peerj.10763#supplemental-information.

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
