# Peer review of "Morphological grounds for the obligate aerial respiration of an aquatic snail: functional and evolutionary perspectives"

_PeerJ, doi:10.7717/peerj.10763_

## Round 0.1 · original submission · Major Revisions

I have heard back from three expert reviewers, with at first view what appear to be widely differing views. However, upon examination, the second reviewer offers a lot of important comments, and thus these could be construed as a major revisions decision. Similarly, the third reviewer does not have issue with your data, but the Discussion, and they recommend a complete reworking of this section, and this could also be construed as a major revisions decision. I can agree with the comments of both reviewers 2 and 3 (and those of reviewer 1), and in summary, there is a lot of work to do. That said, I do not believe it is insurmountable, and my decision is that major revisions are needed. Please consider the comments of particularly reviewers 2 and 3 carefully, and respond in detail to their comments.

I look forward to seeing a revised version of your work.

·

Basic reporting

I am very pleased about the present contribution. Descriptions, preservation of tissues and quality of photos are all of high quality.

Accordingly, all following comments and recommendations are made in the sense to make an already very good paper excellent.

Experimental design

The experimental design, light and electron (SEM, TEM) microscopy, appears adequate to the questions to be studied.

Validity of the findings

The paper continues, widens and summarizes the previous studies by the authors on the same species, most of them being published in PeerJ.

Additional comments

General:
Term „vessel“: I am aware that (unfortunately) this term is regularly used for all molluscan blood tubes. However, a true „vessel“ ist clearly defined by the presence of an endothelium, a condition which - as outlined (line 340ff) - simply is not the case in ampullariids (contrary to e.g., cephalopods or vertebrates). In particular concerning gas transport the difference is crucial, because in case of true vessels an additional cell-layer (the endothelium) needs to be overcome. Accordingly, I recommend to use the widely used term „sinus“ instead of „vessel“ to express the difference (see also Ponder et al. 2019: Biology and Evolution of the Mollusca. Vol. 1: pp. 295-296, CRC Press).

Details:
Line 55: Replace Helicidae by Stylommatophora. You may mention that in case of terrestrial slugs epidermal gas exchange is a major part of respiration.
Line 66: You should add here the detailed anatomical and histological descriptions by Starmühlner (1969: Malacologia 8: 1-434; available via BHL) on Pila cecillei (p.119ff) from Madagascar.
Fig. 1: You may mention that the sections are shown in frontal view. Add bm to the legend.
Fig. 6: Text is somewhat confusing and may cause misunderstandings (see above about „vessels“): Blood chambers are in principle intercellular spaces (micro-sinuses) between the epithelial cells of the lung, the latter do not form an endothelium, however.

Reviewer 2 ·

Basic reporting

English: Recommend active voice
Literature: Appears comprehensive - could better address what elements contribute int he paper
Figures - professional
Hypotheses - none really present - could be better structured around questions

See additional comments below

Experimental design

Research aims could be more clear
Could better identify how this work expands past work and how it fills in an important knowledge gap
Methods seem comprehension but not in a position to evaluate

See below for more comments

Validity of the findings

Novelty could be more clear.
Data images archived.
Conclusions well done - rest of paper could be more succinct.

Additional comments

Morphological grounds for the obligate aerial respiration of an aquatic snail: Functional and evolutionary perspectives
Summary: This paper describes the structure of the lung for Pomacea canaliculata, a species of concern given its ability to tolerate a wide range of environments. The anatomical descriptions appear thorough and the figures professionally rendered.
The MS would benefit from:
1. A better description of how this work advances what has previously been done in the field
2. A broader context for why/how a detailed description of this anatomy and functional morphology adds to our understanding of the ecology and evolution of apple snails.
3. A set of clear objectives or questions that the investigation answered
a. Could complement the “Conclusions” found at the end
4. A synthesis of what this work found in terms of expected, new or different from past studies (i.e. dividing into the main morphological features)
5. Partitioned use of references to refer to specific information or simply cite them chronologically
6. Comparisons of lungs between Pomacea canaliculata and Pomacea maculata – not sure how one can describe as “most developed” without more comparative work. Pomacea maculata lays much larger clutches out of water and may spend more time air breathing than P. canaliculata.
7. Grammatically – chose active voice and avoid use of semicolons and colons. Simplify sentence structure.
Note to Editor: While I do work on apple snails, my experience and strengths do not lie in physiology, morphology or histology.
Abstract:
• Would start with a bigger picture than the focus on a single organism such as the importance of adaptations for amphibiousness
• Little clarity – In this study, we offer supporting evidence (?) versus complimentary?
• Some examples where the language could be improved include increased use of active voice:
o Example: A pavement epithelium lines the lung cavity … instead of “is lined by”

Introduction:
• Could combine sentences – Higher oxygen concentrations and diffusion rates of air, coupled with its lower viscosity, make oxygen update by an animal easier in air than water. Not surprisingly, air-breathing has…
• 58 – “with later development of a lung”
• 66 – “only a few studies examine”
• 69 – how does this work advance the work of Rodriguez
• 77 – this piece of information seems out of place
Methods – outside area of expertise – cannot comment on appropriateness
• Some examples where the language could be improved include increased use of active voice:
o Methods: preferred use of “we” - We obtained samples and stained them….
• 83 – why males? And how many?
Results –
• 145 – 147 – can the authors support this difference in muscle layer thickness quantitatively?
• Figure 2 includes 3 ventral views and only one dorsal view. It would be better if balanced or show an informative diagram for C/D (same view) that appears between A/B (mirror image).
• I would make a table of the anatomical parts and their function
• 223-226 – does not warrant its own paragraph
• 227 – 229 – sounds more like a discussion item
• It seems Figure 10 would be better placed earlier in the manuscript as an overview and then have boxes or arrows that point to the more specific SEM photos
• Figure 11 should be drawn to scale to increase its usefulness

Discussion –
• 292 – usually cite references in chronological order
• 297 – “their lungs” – what is the reference here? Apple snail lungs appear homologous to the pallial cavity of other gastropods? “In these gastropods” – which ones? Hard to discern the meaning behind these sentences.
• 303 – 308 – reads as Results (possibly redundant)
• 311 – 313 – what’s the mechanism for this vessel only appearing in some Pomacea species? What do those species have in common? Do they belong to the same clade? Could expand Discussion of their lower dependence on water with more examples.
• 323 – comma after “through it,” but it also…
• 335 – form and not forms
• 336 – “Our results agree” – worth being specific on which results
• 346 – 349 – Good information but complex sentences makes it a little hard to decipher
• 354 – do the authors mean “phospholipidic”
• 380 – I would disagree that the knowledge of phylogenetic relationships in Ampullariidae only depends on molecular characteristics. I
• 382 – This section on comparative information provides some interesting insights and should be developed further and more clearly connected to the results of this study.
• 394 – end the sentence instead of using a colon. Follow with “For instance,”
• The Discussion overall seems partially redundant and overly extensive
• The subtitles inform the reader as to the subject but not its importance
• 498 – cool that you found something new – the rhogocytes should be given more emphasis – “These cells had not been reported before in the Ampullariidae.”

Reviewer 3 ·

Basic reporting

Rodriguez et al. present a follow up study of the respiratory system of an ampullariid species. Ampullariid snails are easily accessible as they are abundant invading pests and aquarium pets. For this reason their very well investigated also regarding anatomical details such as the respiratory system.
This study provides an extensive description of details of anatomical and histological details surrounding the lung. These results appear valuable as they contain details (at the TEM-level - type of small lacunar system at the base of the epithelium) that may have been overlooked or confused in previous literature.
However, the theoretical treatment of the results (Discussion) is extremely poor. The topics (e.g. comparison with vertebrates, function of rhogozytes, how is the tissue tightened together?) are speculative and/or meaningless. It seems there is a shortage of ideas accompanying the findings, which may due to oversplitting their results (compare Rodriguez et. al, 2019).

Experimental design

The morphological examination has been well done with adequate methodology.

Validity of the findings

Mediocre.

Additional comments

In summary I CLEARLY vote against the publication of this MS UNLESS the theoretical part is FULLY new prepared using different approaches.

For this I recommend to remain within the systematic vicinity of ampullariids and related gastropods (functionally related - pulmonates). A review of previous data of amullariids would be helpful/desirable (tables on data/results). Detailed reasons why previous/other results should be dismissed would be interesting. How does a pulmonate (basommatophoran, sylommatophoran) lung surface compare in detail? Diagrammatic comparative figures could be beneficial. Functional comparisons to other lung breathing gastropods (e.g. lung always air filled or sometimes water filled?) in general could be exploited.

Please find attached the PDF MS with specific comments of mine.

Annotated reviews are not available for download in order to protect the identity of reviewers who chose to remain anonymous.

---

## Round 0.2 · Minor Revisions

Thank you very much for the effort on your revision; all reviewers have noted your work in much improved. There are only some small final details to address from reviewers 1 and 3. I imagine these comments will not take too much time to address, and look forward to seeing a revised version of your manuscript in the near future.

·

Basic reporting

This is a valuable contribution to our understanding of the structure and function of the ampullariid lung serving as a model of terrestralization in gastropods. It continues a high-quality series of papers on the overall subject.

There are only certain minor points which should be considered (see below): .

Experimental design

I am pleased about the various methodologies applied and about the high quality of the LM, SEM and TEM images.

Validity of the findings

The paper provides further progress in our understanding of the model system Pomacea.

Additional comments

This is a valuable contribution to our understanding of the structure and function of the ampullariid lung serving as a model of terrestralization in gastropods. It continues your high-quality series of papers on the overall subject.

I am pleased about the various methodologies applied and about the high quality of the LM, SEM and TEM images.

There are only certain minor points which should be considered: .

Line 55: no – the other way around: originally gastropods (molluscs) had a mantle cavity, and the inner mantle epithelium has been always used for respiration too, the gill (better: ctenidia) are basically used for water ventilation rather than for respiration, in particular in small animals. In case of air-breathing the gills (ctenidia) are no longer necessary and are usually reduced or lost.

Line 278: Better to cite the new textbook of Ponder & Lindberg 2019: Biology and Evolution of the Mollusca. Vol. 1. CRC Press, Taylor & Francis Group, Boca Raton FL, 864 pp

Line 392: hematopoietic

Wells RGM (1980): add „The Institute of Biology´s Studies in Biology 127, ISBN 10: 9780713128062, 71 pp"

Fig. 7ff: the labeled “hemocytes” (hcy) may be specified as amoebocytes with their typical pseudopodia.

Table 1: Haszprunar (1996) instead of “(1995)”

Reviewer 3 ·

Basic reporting

The MS is sufficiently improved. The discussion appears to be completely reworked, it is clear and interesting and does not contain unfounded consideration anymore in this version.
It is generally well written now.

Experimental design

Well done.

Validity of the findings

Interesting in the field of invertebrate biology.

Additional comments

I placed a few comments and suggestions in the PDF MS.

Annotated reviews are not available for download in order to protect the identity of reviewers who chose to remain anonymous.

---

## Round 0.3 · accepted · Accept

Thank you for your efforts in doing this revision - I am now happy to accept this manuscript and move it into production. Congratulations!